# Biological Screening of Novel Structural Analog of Celecoxib as Potential Anti-Inflammatory and Analgesic Agent

**DOI:** 10.3390/medicina55040093

**Published:** 2019-04-05

**Authors:** Hristina Zlatanova, Stanislava Vladimirova, Ilia Kostadinov, Delian Delev, Tanya Deneva, Ivanka Kostadinova

**Affiliations:** 1Department of Pharmacology and Clinical Pharmacology, Faculty of Medicine, Medical University Plovdiv, 4002 Plovdiv, Bulgaria; ilia_197575@abv.bg (I.K.); delevg@gmail.com (D.D.); gulabche@yahoo.com (I.K.); 2Department of Organic Synthesis and Fuels, Faculty of Chemical Technologies, University of Chemical Technology and Metallurgy, 1756 Sofia, Bulgaria; vladimirova.s@mail.bg; 3Department of Clinical Laboratory, Faculty of Pharmacy, Medical University Plovdiv, 4002 Plovdiv, Bulgaria; tdeneva@mail.bg

**Keywords:** NSAIDs, pyrrole, celecoxib, carrageenan, formalin, tail-flick, plantar test

## Abstract

*Background and objectives*: The clinical use of non-steroidal anti-inflammatory drugs is limited due to high incidence of adverse drug reactions. The pyrrole heterocycle is included in the chemical structure of a number of drugs with various activities and shows relatively good tolerability and safety. The objectives of our study were to evaluate the analgesic and anti-inflammatory activity, as well as possible organ toxicity, of 2-[3-acetyl-5-(4-chloro-phenyl)-2-methyl-pyrrol-1-yl]-3-(1H-indol-3-yl)-propionic acid (compound 3g), a novel N-pyrrolylcarboxylic acid structurally similar to celecoxib. *Materials and methods*: All experiments were performed on 6-week-old male Wistar rats divided into parallel groups (n = 8). Antinociception was assessed using animal pain models with thermal and chemical stimuli (paw withdrawal, tail-flick, and formalin tests). Criteria for the analgesic effect were increased latency in the paw withdrawal and tail-flick tests and decreased paw licking time in the formalin test compared to animals treated with saline (control). Anti-inflammatory activity was measured using a carrageenan-induced paw edema model; the criterion for anti-inflammatory effect was decreased edema compared to control. Blood samples were obtained after animals were sacrificed to assess possible organ toxicity. Statistical analysis was performed with IBM SPSS 20.0. *Results*: 2-[3-Acetyl-5-(4-chloro-phenyl)-2-methyl-pyrrol-1-yl]-3-(1H-indol-3-yl)-propionic acid had analgesic action against chemical stimulus after single and multiple administration and against thermal stimulus after single administration. Compound 3g significantly suppressed carrageenan-induced paw edema after both single and continuous administration. After continuous administration, hematological tests showed that compound 3g decreased leukocyte and platelet levels and elevated serum creatinine levels. *Conclusions*: Antinociception with the tested compound is most likely mediated by spinal, peripheral, and anti-inflammatory mechanisms. Possible tolerance of the analgesic action at the spinal level develops after continuous administration. Anti-inflammatory activity is significant and probably the leading cause of antinociception. After multiple administration, compound 3g showed signs of potential nephrotoxicity and antiplatelet activity, as well as suppression of leukocyte levels.

## 1. Introduction

Diagnosis and treatment of acute and chronic pain is a medical and social problem. Inadequate pain control can lead to impaired rehabilitation of patients and prolonged hospitalization [1]. Treatment of pain is an unsolved problem that poses many questions for clinicians and experimentalists.

Persisting inflammatory stimuli or deregulation of the resolution phase mechanisms cause chronic inflammation, acknowledged as a key factor in the evolution of a number of diseases [2]. Nonsteroidal anti-inflammatory drugs (NSAIDs) are among the few drug groups used to reduce inflammation and treat chronic inflammatory disorders [3]. The clinical use of NSAIDs is highly limited, because 20–25% of all recorded adverse drug reactions (ADRs) are due to their use [4], including gastrointestinal complications [5], hepatotoxicity [6], nephrotoxicity [7], and bone marrow suppression. Cyclooxygenase-2 (COX-2) inhibitors were created in an attempt to minimize these side effects, although they show serious side effects of their own, such as prothrombotic state and increased risk of myocardial infarction and stroke [8,9].

The search for new drugs with analgesic and anti-inflammatory activity that have improved tolerability and safety and high efficiency is a necessity. One approach to the design and synthesis of novel drug substances is to base the new molecule on the architecture of an existing medication [10].

The pyrrole heterocycle is included in the chemical structure of a number of drugs with various activities: Antimycobacterial, antifungal, dipeptidyl peptidase 4 inhibition, and others [11,12,13]. Among its diverse biological activities, it has relatively good tolerability and safety [14]. Its anti-inflammatory properties have been confirmed in numerous recent studies [15,16,17,18]. The newly synthesized derivative discussed in the present study, 2-[3-acetyl-5-(4-chloro-phenyl)-2-methyl-pyrrol-1-yl]-3-(1H-indol-3-yl)-propionic acid, or compound 3g, is a derivative of N-pyrrolylcarboxylic acid and its structure is oriented to that of the modern drug celecoxib (a selective COX-2 inhibitor from the group of coxibs). Its chemistry, design, synthesis, and characterization by spectroscopy and thin layer chromatography are described by Vladimirova and Bijev [19]. The structure of the novel compound and its similarity to celecoxib are demonstrated in Figure 1.

The aim of this study is to evaluate the antinociceptive and anti-inflammatory properties of compound 3g after single and continuous (14-day) administration using animal models of pain and inflammation and to assess the organ toxicity of the substance after continuous (14-day) treatment.

## 2. Materials and Methods

The experiment was approved by the Ethics Committee on Animals of the Bulgarian Food Safety Agency with permit no. 128/09.12.2015 and by the decision of the Ethics Committee at the Medical University of Plovdiv, protocol no. 2/31.03.2016.

### 2.1. Reagents 

Reagents used in the experiments were as follows: Metamizole sodium amp. 500 mg/mL 2 mL (Sopharma AD, Sofia, Bulgaria), diclofenac sodium amp. 75 mg/3 mL (Hexal AG, Holzkirchen, Germany), NaCl 0.9% (Sopharma AD, Sofia, Bulgaria), lambda-carrageenan (Merck, Darmstadt, Germany), formalin 0.2% (Merck, Darmstadt, Germany), and 2-[3-acetyl-5-(4-chloro-phenyl)-2-methyl-pyrrol-1-yl]-3-(1H-indol-3-yl)-propionic acid (compound 3g). All substances were dissolved in saline and administered intraperitoneally. The doses used in the experiment were determined by acute oral toxicity tests.

### 2.2. Experimental Animals 

All experiments were performed on 6-week-old male Wistar rats (*n* = 48) weighing 150 ± 20 g, randomly divided into 6 parallel experimental groups, as described in Table 1. The animals were maintained on a light/dark cycle of 12/12 h in a temperature-controlled environment with food and water available ad libitum.

### 2.3. Antinociception Assessment

#### 2.3.1. Plantar Test (Hargreaves Method) 

Each animal was placed in an individual compartment and left unrestrained. After an acclimation period, an infrared heat source (Ugo Basile, Italy) was positioned under the glass floor directly beneath the hind paw of the animal, and paw withdrawal latency (in seconds) was recorded automatically. Cutoff time was set at 30 s to avoid unnecessary overheating of the paw [20].

The tests were performed 1, 2, and 3 h after intraperitoneal administration of saline solution, metamizole sodium, or compound 3g in doses of 10, 20, and 40 mg/kg body weight (b.w.) for the respective experimental groups. The criterion for analgesic action was increased reaction time compared to animals treated with saline.

#### 2.3.2. Tail-Flick Test 

The rat was held by the operator, and a radiant heat source (Ugo Basile, Italy) was focused on the underside of the tail approximately 3 cm from its distal end. The time in seconds for tail withdrawal was automatically recorded. Cutoff time was set at 15 s to avoid tissue damage [21,22].

The tests were performed 1, 2, and 3 h after intraperitoneal administration of saline solution, metamizole sodium, or compound 3g in doses of 10, 20, and 40 mg/kg body weight (b.w.) for the respective experimental groups. The criterion for analgesic action was increased reaction time compared to animals treated with saline.

#### 2.3.3. Formalin Test 

For each animal, 0.2% 200 µL formalin was injected intradermally in a hind paw 1 h after administration of the tested substance [23]. The cumulative time spent licking/biting the injected paw was measured (in seconds) during the first 10 min (1st phase) and between the 20–30 min (2nd phase) after administration of formalin. The criterion for analgesic effect was decreased paw licking time in comparison to the control group [24].

### 2.4. Anti-Inflammatory Activity Assessment

The volume of the right hind paw of each animal of all experimental groups was measured with a plethysmometer (Ugo Basile, Italy) prior to treatment. Edema was induced by injection of 0.1 mL of a 1% suspension of carrageenan in 0.9% saline solution into the right hind paw [25]. Saline solution, diclofenac sodium, or compound 3g (10, 20, and 40 mg/kg b.w.) was administered intraperitoneally immediately after the injection of carrageenan. Paw volume was measured 2, 3, and 4 h after administration of carrageenan. The percentage of paw edema was calculated using the following formula:(1)$$\mathrm{Percentage}\;\mathrm{of}\;\mathrm{paw}\;\mathrm{edema}=\frac{Vt-Vo}{Vo}\times 100$$
where *Vo* is mean paw volume at 0 h and *Vt* is mean paw volume at a particular time interval.

The criterion for anti-inflammatory effect was decreased paw edema compared to the control group [26].

### 2.5. In Vitro Testing of Hematological and Biochemical Parameters

Animals were sacrificed by cervical dislocation after continuous (14-day) treatment with compound 3g (10, 20, and 40 mg/kg b.w.) and exsanguinated; blood was collected in 2 vials. Analysis of hematological parameters (red blood cells, white blood cells, platelets, and hemoglobin) was performed using an ADVIA 2120i automated hematology analyzer (Siemens Diagnostic). Serum was separated after coagulating at 37 °C for 60 min and centrifuged at 3000 rpm for 10 min. Serum was used for the estimation of aspartate aminotransferase (AST), alanine aminotransferase (ALT), creatinine, urea, and blood glucose. All clinical chemical analyses were carried out on an Olympus AV 480 Analyzer (Beckman Coulter) according to the manufacturer’s instructions.

### 2.6. Statistical Evaluation

Statistical analysis of the obtained data was performed with IBM SPSS 20.0 software, using ANOVA, Tukey post hoc when equal variances were assumed, and Games–Howell post hoc when equal variances were not assumed. The normality of distribution was established with Shapiro–Wilk test. Results are expressed as arithmetic mean and standard error of the mean (mean ± SEM). *p* value ≤ 0.05 is considered statistically significant. Results are summarized in tables.

## 3. Results

### 3.1. Antinociception Assessment

Based on one-way ANOVA testing, in general, a statistically significant difference exists among the experimental groups. In the plantar and formalin tests, this difference is observed at all hours and phases, respectively, after both single and multiple administrations. In the tail-flick test such difference is observed only at the first two hours of testing after single and multiple administrations. Where statistically significant differences are observed, multiple comparisons with post-hoc testing are performed to determine the specific groups that show this significance.

The reference analgesic metamizole sodium [27] showed significant analgesic effect in all tests after both single and multiple administration.

In the plantar test (Table 2), compound 3g in any of the tested doses did not change paw withdrawal time compared to the animals treated with saline. Continuous (14-day) administration of compound 3g in doses of 10, 20, and 40 mg/kg b.w. did not affect paw withdrawal latency compared to the control animals. The reflex response observed during the paw withdrawal test was mediated by supraspinal pathways [21], hence we can assume that such circuits are not part of the antinociception induced by compound 3g.

In the tail-flick test (Table 3), compound 3g in doses of 10, 20, and 40 mg/kg b.w. significantly increased tail withdrawal time at 2 h compared to the control group. After repeated administration, compound 3g in doses of 10, 20, and 40 mg/kg b.w. did not affect reaction time compared to the animals treated with saline. 

In the formalin test (Table 4), compound 3g in doses of 10, 20, and 40 mg/kg b.w. significantly decreased the time spent licking/biting the paw in both phases of the test compared to the animals treated with saline. After repeated administration, compound 3g in doses of 10, 20, and 40 mg/kg b.w. significantly decreased paw licking/biting time in the first and second phases of the test compared to the control animals. 

### 3.2. Anti-inflammatory Activity Assessment

Statistically significant difference exists between the experimental groups at all hours after both single and multiple administrations of the tested substances. Multiple comparisons with post-hoc testing are performed to determine the specific groups that show this significance.

In the carrageenan model of inflammation (Table 5), diclofenac sodium [18], used as reference substance with anti-inflammatory effect, significantly reduced paw edema at 2, 3, and 4 h compared with the control group. After single administration, compound 3g at a dose of 10 mg/kg b.w. significantly reduced paw edema at 2 h compared with the animals treated with saline. A 20 mg/kg b.w. dose reliably suppressed carrageenan-induced edema at 2 and 4 h compared to the control group. A 40 mg/kg b.w. dose significantly inhibited paw edema at all tested hours compared to the animals treated with saline. After repeated administration, compound 3g at doses of 10, 20, and 40 mg/kg b.w. significantly reduced carrageenan-induced edema at 2, 3, and 4 h compared to animals treated with saline. 

### 3.3. Hematological and Biochemical Parameters

Statistically significant difference exists between the experimental groups concerning the following biochemical and hematological parameters: ALT, AST, urea, creatinine, glucose, WBC, PLT. Multiple comparisons with post-hoc testing are performed to determine the specific groups that show this significance.

Compound 3g at doses of 10, 20, and 40 mg/kg b.w. reliably reduced ALT and urea values and significantly increased creatinine values compared to the control group. Compound 3g at doses of 10 and 20 mg/kg b.w. significantly reduced plasma glucose levels compared to the animals treated with saline. A 20 mg/kg b.w. dose significantly reduced AST levels compared to the control group. Compound 3g did not affect levels of red blood cells or hemoglobin compared to animals treated with saline. Compound 3g at doses of 10, 20, and 40 mg/kg b.w. reliably reduced the number of leukocytes in plasma compared to control animals. Doses of 20 and 40 mg/kg b.w. significantly reduced the number of platelets compared to the control group (Table 6 and Table 7).

## 4. Discussion

The results in our study show that in experimental conditions 2-[3-acetyl-5-(4-chloro-phenyl)-2-methyl-pyrrol-1-yl]-3-(1H-indol-3-yl)-propionic acid has significant anti-inflammatory activity and shows analgesic action against chemical and thermal stimuli. These effects are registered after both single and multiple administrations of the compound with the exception of one of the pain models where we observe loss of analgesic effect after continuous administration.

Our study is, of course, not without limitations. The number of animals used, even though chosen after consultation with a statistician, is still a relatively small sample. The experimental substances are tested only on rats and other animal species may differ in their sensitivity to the compounds. We tried to limit the human factor in our experimental procedures ergo the choice of tail-flick and plantar tests where measurement is automatic. Still, in the tail-flick test a handler holds the animal immobile and in the formalin test the recording of seconds spent biting/licking their paw is manual. Our choice of doses also has limitations. While based on acute toxicity studies, the usage of only three increasing doses makes it difficult to observe dose-dependency of the effect with a high degree of certainty; higher doses might show a more pronounced pharmacological effect. 

The tail-flick test is considered precise, since it is measured electronically, and there is a small inter-animal variability in reaction times [22]. The observed reflex response in this test occurred at the spinal level [21], therefore we suggest that spinal pathways are implicated in the analgesic mode of action of compound 3g. After continuous administration, the analgesic action was lost. This could indicate possible development of tolerance to the analgesic effect at the spinal level. Since tolerance, antinociception at the spinal level, and increased latency in the tail-flick test are very typical of opioids [28], we propose that the endogenous opioidergic system is involved in the analgesia induced by compound 3g. Involvement of this system in the antinociception caused by NSAIDs is supported by data from recent studies, which propose that gamma-aminobutyric acid (GABA)-containing synapses act as sites where NSAIDs converge with endogenous opioids [29]. Multiple intraperitoneal administration of non-opioid analgesics in rats induces antinociception and causes tolerance, as well as cross-tolerance to morphine. Administration of naloxone (a specific µ-receptor opioid antagonist) significantly decreases the antinociceptive effects of NSAIDs [29]. Further investigations of possible involvement of opioidergic mechanisms are necessary to establish the specific mechanisms of antinociceptive action of compound 3g.

Intraplantar administration of formalin leads to biphasic pain response. The initial phase is the result of direct activation of peripheral nociceptors and lasts approximately 10 min. The second phase begins after a 10-minute quiescent period and is due to the release of local endogenous pro-inflammatory mediators, causing peripheral inflammatory processes and subsequent sensitization of nociceptive spinal neurons [24]. In our study, compound 3g, after both single and multiple administration, showed significant antinociceptive effects in both phases of the formalin test. We can speculate that peripheral mechanisms are part of the mode of antinociceptive action of compound 3g, which would be in line with the cyclooxygenase-inhibiting activity expected of this substance. The analgesia produced by the compound during the second phase of the test could result from either suppression of pro-inflammatory mediators or modulation of pain transmission at the spinal level. After single administration, compound 3g increased latency in the tail-flick test, a sign that antinociception at the spinal level does occur. However, this effect was lost after continuous administration, and since the compound continued to show analgesic effect in the second phase of the formalin test, we suggest that inflammation suppression plays a major role in the antinociception induced by compound 3g.

Carrageenan-induced inflammation shows two distinct phases, vascular and cellular [30]. Bradykinin, serotonin, histamine, and several pro-inflammatory cytokines (tumor necrosis factor α, interleukin-1β) are responsible for the vasodilation and initial extravasation during the first couple of hours. Prostaglandins play a major role in the cellular phase, which occurs approximately four hours after intraplantar administration of carrageenan [25]. Suppression of this last phase is typical of NSAIDs and correlates to a great extent with their therapeutic efficacy [31]. This makes the model particularly suitable for registering the anti-inflammatory effects of novel substances with a suspected COX-inhibiting mode of action. After single administration, the highest dose used, 40 mg/kg b.w., reduced edema at all tested hours. After multiple administration, all doses of compound 3g significantly inhibited paw edema at 2, 3, and 4 h. This suggests the need for a higher concentration and possibly for accumulation of the compound in order for it to bind stably to COX. Inhibition of paw edema was especially pronounced during the first two hours, which suggests that compound 3g influences the vascular phase of inflammation.

In controlled clinical studies, patients treated with celecoxib presented with elevated liver enzymes [32]. We decided to screen the novel substance for possible hepatotoxicity by evaluating the levels of liver transaminases (AST and ALT) as markers for liver damage. The groups treated with compound 3g for 14 days presented with decreased levels of liver enzymes compared to controls. Nephrotoxicity can also be observed during treatment with NSAIDs. Renal damage can be present with interstitial nephropathy and tubulointerstitial nephritis, although kidney injury is usually reversible and patients recover after cessation of NSAID use [33,34]. Nephrotoxicity is rare in celecoxib use, though elevated levels of blood urea and creatinine have been observed [32]. Compound 3g increases creatinine levels while lowering blood urea. Both are freely filtered by the glomerulus, but while urea gets reabsorbed in the tubules, creatinine reabsorption is minimal. Therefore, the increase in creatinine levels could indicate glomerular damage. Urea also reflects functioning of the liver, since its production occurs primarily in the liver. Compound 3g decreases not only urea levels, but also AST and ALT levels, and we theorize that it suppresses liver function. Plasma glucose is also tested as a part of the standard biochemical blood profile. Compound 3g significantly decreased blood sugar levels compared to the animals treated with saline. Studies show that some NSAIDs (aspirin, nimesulide, phenylbutazone) lower blood sugar levels due to the inhibition of glucose transport and enzymatic changes in the intestines [35]. In vitro studies have shown, however, that celecoxib does not significantly affect intestinal permeability [36].

Hematological tests showed the following results: Compound 3g did not affect red blood cells and hemoglobin levels, while it decreased platelets and white blood cells. Platelets are very sensitive to cyclooxygenase inhibitors, since they cannot resynthesize the enzyme. Even NSAIDs that bind reversibly to COX cause a transient decrease in platelet levels and aggregation, although these effects subside after most of the drug is eliminated from the organism [37,38]. For selective COX-2 inhibitors, however, a prothrombotic state is typical, because they block the production of prostacyclin (PGI_2_) but not thromboxane A2 (since TxA_2_ synthesis depends on preserved COX-1 activity) [8]. Suppression of platelets with compound 3g could mean that either it does not possess the expected selectivity toward COX-2 or platelet inhibition results from another mode of action. Bone marrow suppression and agranulocytosis, in particular, are not typical adverse drug reactions to NSAIDs. Metamizole is one of the few COX inhibitors that show these reactions [39], although it does not belong to the group of NSAIDs, but rather to the group of analgesics–antipyretics. The decrease in white blood cells observed with compound 3g could be due to bone marrow suppression. Another possible explanation is leftover anti-inflammatory action, since blood samples were obtained approximately 24 h after injection of carrageenan. This could potentially explain the significant difference between white blood cell levels in the control group and groups treated with compound 3g.

## 5. Conclusions

Compound 3g (2-[3-acetyl-5-(4-chloro-phenyl)-2-methyl-pyrrol-1-yl]-3-(1H-indol-3-yl)-propionic acid) induces an antinociceptive effect after single and continuous (14-day) administration in rats. Antinociception with the tested compound is most likely mediated by spinal, peripheral, and anti-inflammatory mechanisms. Regulation of pain transmission with compound 3g does not occur on the supraspinal level. Possible tolerance to the analgesic action develops at the spinal level after continuous administration. 

Compound 3g shows significant dose-dependent anti-inflammatory activity, inhibiting both vascular and cellular phases of inflammation after single and multiple (14-day) administration in rats. 

After continuous administration, compound 3g lowers serum levels of liver transaminases, plasma glucose, and blood urea, which suggests suppressed liver function. The compound elevates creatinine levels and shows signs of potential nephrotoxicity.

Compound 3g does not affect hemoglobin and red blood cell levels. It suppresses platelets, which is a typical ADR for most NSAIDs; the compound also lowers white blood cell levels after 14-day administration, although this effect could be partially explained by leftover anti-inflammatory activity.

## Figures and Tables

**Figure 1 medicina-55-00093-f001:**
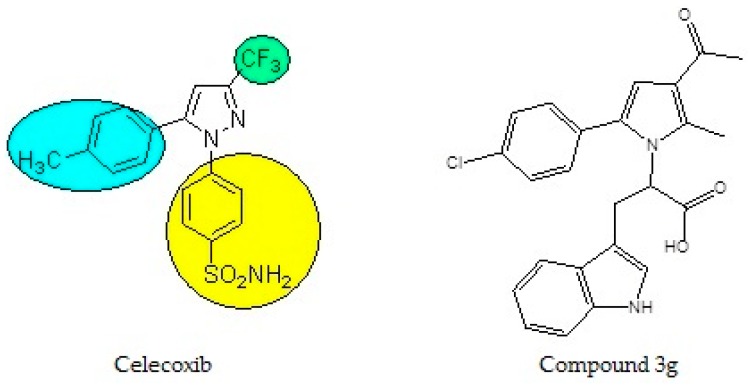
Chemical structure of celecoxib and compound 3g.

**Table 1 medicina-55-00093-t001:** Experimental animal groups.

Experimental Group	Tested Substance	Number of Animals
1	Saline solution	8
2	Metamizole 200 mg/kg b.w.	8
3	Diclofenac 25 mg/kg b.w.	8
4	Compound 3g 10 mg/kg b.w.	8
5	Compound 3g 20 mg/kg b.w.	8
6	Compound 3g 40 mg/kg b.w.	8

Note: mg/kg b.w., milligrams per kilogram body weight.

**Table 2 medicina-55-00093-t002:** Comparison of withdrawal latency (in seconds) in plantar test between control group and groups treated with metamizole and compound 3g in doses of 10, 20, and 40 mg/kg b.w.

Group	Hour	Mean ± SEM	*p*	Mean_1_ ± SEM_1_	*p*
Control	1st	6.75 ± 0.48	-	7.13 ± 1.42	-
	2nd	8.67 ± 2.34		7.70 ± 1.72	
	3rd	10.00 ± 2.10		7.17 ± 2.22	
Metamizole	1st	17.11 ± 1.97	0.002 *&	19.36 ± 2.06	0.001 *#
	2nd	17.10 ± 1.22	0.007 *&	16.30 ± 2.66	0.028 *&
	3rd	18.19 ± 2.24	0.023 *#	13.28 ± 1.37	0.030 *#
3g 10 mg/kg	1st	11.83 ± 2.59	0.09	14.22 ± 1.18	0.003
	2nd	13.73 ± 2.56	0.18	10.80 ± 1.13	0.16
	3rd	14.41 ± 1.01	0.06	9.37 ± 1.10	0.40
3g 20 mg/kg	1st	10.13 ± 1.65	0.10	10.23 ± 1.69	0.19
	2nd	8.52 ± 1.24	0.96	11.28 ± 1.22	0.12
	3rd	14.67 ± 1.27	0.09	9.67 ± 1.89	0.41
3g 40 mg/kg	1st	10.54 ± 1.71	0.07	12.32 ± 2.05	0.06
	2nd	12.51 ± 0.62	0.17	11.77 ± 1.47	0.10
	3rd	13.43 ± 1.07	0.14	8.50 ± 1.90	0.66

Note: mean ± SEM, *p*-values after single administration; mean_1_ ± SEM_1_, *p*-values after multiple (14-day) administration; * *p* < 0.05 compared to control; # Tukey post-hoc was used; & Games–Howell post hoc was used.

**Table 3 medicina-55-00093-t003:** Comparison of withdrawal latency (in seconds) in tail-flick test between control group and groups treated with metamizole and compound 3g in doses of 10, 20, and 40 mg/kg b.w.

Group	Hour	Mean ± SEM	*p*	Mean_1_ ± SEM_1_	*p*
Control	1st	3.33 ± 0.22	-	3.50 ± 0.70	-
	2nd	2.58 ± 0.10		2.70 ± 0.25	
	3rd	3.17 ± 0.45		3.93 ± 0.48	
Metamizole	1st	5.14 ± 0.70	0.042 *&	5.80 ± 0.67	0.039 *#
	2nd	3.79 ± 0.39	0.018 *#	4.42 ± 0.65	0.033 *#
	3rd	4.19 ± 0.79	0.31	4.32 ± 0.97	0.73
3g 10 mg/kg	1st	3.81 ± 0.30	0.25	3.75 ± 0.65	0.80
	2nd	3.33 ± 0.23	0.016 *#	3.48 ± 0.43	0.14
	3rd	3.26 ± 0.16	0.83	2.83 ± 0.27	0.07
3g 20 mg/kg	1st	3.64 ± 0.32	0.47	3.55 ± 0.56	0.96
	2nd	3.51 ± 0.29	0.019 *#	2.79 ± 0.26	0.82
	3rd	3.75 ± 0.37	0.33	3.06 ± 0.38	0.18
3g 40 mg/kg	1st	3.50 ± 0.35	0.72	3.91 ± 0.41	0.61
	2nd	3.94 ± 0.42	0.015 *#	3.14 ± 0.30	0.29
	3rd	3.15 ± 0.32	0.98	2.96 ± 0.12	0.06

Note: mean ± SEM, *p*-values after single administration; mean_1_ ± SEM_1_, *p*-values after multiple (14-day) administration; * *p* < 0.05 compared to control; # Tukey post-hoc was used; & Games–Howell post hoc was used.

**Table 4 medicina-55-00093-t004:** Comparison of the time spent licking/biting paw (in seconds) in formalin test between control group and groups treated with metamizole and compound 3g in doses of 10, 20, and 40 mg/kg b.w.

Group	Phase	Mean ± SEM	*p*	Mean_1_ ± SEM_1_	*p*
Control	1st	70.17 ± 16.81	-	42.00 ± 9.36	-
	2nd	47.83 ± 11.12		35.00 ± 5.22	
Metamizole	1st	23.83 ± 4.35	0.039 *&	7.38 ± 3.05	0.002 *#
	2nd	18.83 ± 4.07	0.048 *&	11.25 ± 3.90	0.003 *#
3g 10 mg/kg	1st	16.14 ± 2.58	0.023 *&	21.38 ± 4.06	0.046 *#
	2nd	17.29 ± 4.91	0.041 *&	17.13 ± 3.41	0.011 *#
3g 20 mg/kg	1st	12.29 ± 3.83	0.017 *&	15.88 ± 3.17	0.012 *#
	2nd	12.43 ± 5.45	0.012 *&	16.00 ± 3.16	0.006 *#
3g 40 mg/kg	1st	4.83 ± 2.68	0.011 *&	11.00 ± 2.52	0.010 *#
	2nd	11.17 ± 4.50	0.02 *&	18.83 ± 3.19	0.025 *#

Note: mean ± SEM, *p*-values after single administration; mean_1_ ± SEM_1_, *p*-values after multiple (14-day) administration; * *p* < 0.05 compared to control; # Tukey post-hoc was used; & Games-Howell post hoc was used.

**Table 5 medicina-55-00093-t005:** Comparison of percentage of edema in carrageenan-induced paw edema test between control group and groups treated with diclofenac and compound 3g in doses of 10, 20, and 40 mg/kg b.w.

Group	Hour	Mean ± SEM	*p*	Mean_1_ ± SEM_1_	*p*
Control	2nd	47.08 ± 4.06	-	45.77 ± 3.79	-
	3rd	35.70 ± 6.88		43.67 ± 3.88	
	4th	41.55 ± 3.75		49.24 ± 4.33	
Diclofenac	2nd	21.86 ± 2.33	<0.001 *#	14.46 ± 0.89	<0.001 *#
	3rd	14.41 ± 2.93	0.011 *#	11.19 ± 1.39	<0.001 *#
	4th	16.54 ± 3.02	<0.001 *#	7.70 ± 1.25	<0.001 *#
3g 10 mg/kg	2nd	29.25 ± 3.72	0.007 *#	11.64 ± 2.65	<0.001 *#
	3rd	36.74 ± 6.71	0.92	21.31 ± 4.15	0.002 *#
	4th	32.17 ± 4.14	0.12	33.03 ± 4.36	0.024 *#
3g 20 mg/kg	2nd	31.78 ± 2.09	0.006 *#	11.19 ± 4.79	<0.001 *#
	3rd	25.95 ± 5.55	0.29	13.93 ± 4.59	<0.001 *#
	4th	27.71 ± 5.14	0.050 *	22.85 ± 3.68	0.001 *#
3g 40 mg/kg	2nd	20.09 ± 3.96	<0.001 *#	7.21 ± 2.01	<0.001 *#
	3rd	18.41 ± 3.76	0.048 *	13.62 ± 1.59	<0.001 *#
	4th	14.04 ± 3.82	<0.001 *#	19.50 ± 3.33	<0.001 *#

Note: mean ± SEM, *p*-values after single administration; mean_1_ ± SEM_1_, *p*-values after multiple (14-day) administration; * *p* < 0.05 compared to control; # Tukey post-hoc was used.

**Table 6 medicina-55-00093-t006:** Comparison of biochemical parameters between control group and groups treated with compound 3g in doses of 10, 20, and 40 mg/kg b.w.

Group	Biochemical Parameter	Mean ± SEM	*p*
Control	AST	217.63 ± 8.92	-
	ALT	63.50 ± 5.78	
	CREA	37.25 ± 0.94	
	UREA	6.45 ± 0.26	
	GLUC	10.18 ± 0.22	
3g 10 mg/kg	AST	202.25 ± 14.72	0.390
	ALT	36.38 ± 5.80	0.005 *#
	CREA	44.88 ± 1.64	0.001 *#
	UREA	5.09 ± 0.25	0.002 *#
	GLUC	8.41 ± 0.57	0.017 *#
3g 20 mg/kg	AST	173.88 ± 7.37	0.002 *#
	ALT	37.38 ± 2.27	0.002 *#
	CREA	43.00 ± 1.00	0.001 *#
	UREA	4.61 ± 0.35	0.001 *#
	GLUC	8.94 ± 0.27	0.003 *#
3g 40 mg/kg	AST	218.13 ± 7.01	0.970
	ALT	38.13 ± 3.31	0.002 *#
	CREA	43.50 ± 1.30	0.002 *#
	UREA	4.56 ± 0.22	<0.001 *#
	GLUC	9.48 ± 0.38	0.13

Note: mean ± SEM, *p*-values after multiple (14-day) administration; AST, aspartate aminotransferase; ALT, alanine aminotransferase; CREA, creatinine; UREA, urea; GLUC, glucose; * *p* < 0.05 compared to control; # Tukey post-hoc was used.

**Table 7 medicina-55-00093-t007:** Comparison of hematological parameters between control group and groups treated with compound 3g in doses of 10, 20, and 40 mg/kg b.w.

Group	Hematological Parameter	Mean ± SEM	*p*
Control	RBC	6.33 ± 0.25	-
	HGB	124.50 ± 4.69	
	WBC	10.78 ± 0.78	
	PLT	659.75 ± 70.52	
3g 10 mg/kg	RBC	6.63 ± 0.17	0.400
	HGB	130.00 ± 4.22	0.440
	WBC	4.86 ± 0.74	<0.001 *#
	PLT	387.00 ± 166.59	0.180
3g 20 mg/kg	RBC	6.70 ± 0.13	0.210
	HGB	133.50 ± 2.85	0.120
	WBC	3.54 ± 0.61	<0.001 *#
	PLT	339.00 ± 63.26	0.011 *#
3g 40 mg/kg	RBC	6.84 ± 0.36	0.260
	HGB	129.63 ± 1.83	0.340
	WBC	3.27 ± 0.70	<0.001 *#
	PLT	348.50 ± 53.05	0.006 *#

Note: mean ± SEM, *p*-values after multiple (14-day) administration; RBC, red blood cell; HGB, hemoglobin; WBC, white blood cell; PLT, platelet; * *p* < 0.05 compared to control; # Tukey post-hoc was used.

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
