# Peer review of "Biological Screening of Novel Structural Analog of Celecoxib as Potential Anti-Inflammatory and Analgesic Agent"

_medicina, 2019, doi:10.3390/medicina55040093_

Round 1
Reviewer 1 Report
Please give structures of all the drugs studied and compare them with the conventionnal NSAIDs.
Author Response
Response to Reviewer 1 Comments
Point 1: Please give structures of all the drugs studied and compare them with the conventional NSAIDs.
Response 1: The chemical structure of the studied compound 3g has been presented in a figure, as well as Celecoxib’s structure, to which 3g is oriented.
Reviewer 2 Report
Comments to the Author
I have the pleasure to review this Manuscript ” Biological screening of novel structural analog of Celecoxib as potential anti-inflammatory and analgesic agent.
This manuscript seems interesting and promising results in analgesic and anti-inflammatory activity, as well as possible organ toxicity of 2-[3-Acetyl-5-(4-chloro-phenyl)-2-methyl-19 pyrrol-1-yl]-3-(1H-indol-3-yl)-propionic acid (compound 3g), a novel N-pyrrolylcarboxylic acid, structurally similar to Celecoxib.. This study described experiments which are performed on 6-weeks 21 old male Wistar rats divided in parallel groups of n=8. Antinociception was assessed by using animal pain models with thermal and chemical stimuli. Analgesic effect was tested with increased latency in paw withdrawal and tail-flick test and decreased paw licking time in formalin test, compared to animals treated with saline (control). Anti-inflammatory activity was measured using carrageenan-induced paw edema model; criterion for anti-inflammatory effect was decreased edema compared to control. Also blood samples were obtained after animals are sacrificed to assess possible organ toxicity. Statistical analysis is performed with IBM SPSS 20.0. This new substans suggests suppressed liver functions and shows signs of potential nephrotoxicity. It suppresses platelets Compound 3g does not affect hemoglobin and red blood cell levels. This is a very well written manuscript that examined a new structural analog of celecoxib. However, this is new and innovating and the question is relevant in daily practice.
Specific comments
No comments
Author Response
Response to Reviewer 2 Comments
Point 1: Extensive editing of English language and style required.
Response 1: The manuscript has been sent to MDPI English editing services and has undergone language improvement.
Reviewer 3 Report
In the present paper the compound 2-[3-Acetyl-5-(4-chloro-phenyl)-2-methyl-19 pyrrol-1-yl]-3-(1H-indol-3-yl)-propionic acid (3g), described in a previous paper in 2014, was assayed in different pain models and in an antinflammatory model to evaluated its significance as analgesic and anti-inflammatory drug.
Although the paper is well articulated and the experimental models well described, it has a reduced novelty value compared to the Celecoxib, chosen as reference compound. Furthermore, the considerable adverse effects hypothesized by the authors through the in vitro assays on hematological and biochemical parameters make the compound unattractive for its clinical use.
Anyway, I recommend a minor revision of the paper.
Line 100: introduce doses of compound 3g
Line 101: action is the increase…
Line 102: “…formalin is injected intradermally in one of the hind paws one hour after the administration of the tested substances.… : The authors explain the choice of this time distance
Line 112: instead in Anti-inflammatory activity assay, the authors said that “Saline solution, diclofenac sodium and compound 3g (10, 20 and 40 mg/kg b.w.) are administered intraperitoneally immediately after the injection of carrageenan" without allowing a minimum distance between the test substance and the inflammatory agent. The authors explain this choice.
Moreover, the authors explain because the carrageenan solution was used in 1% instead of 2%.
Finally, the tables 2, 3 and 4 could be replaced with graphs.
Author Response
Response to Reviewer 3 Comments
Point 1: Line 100: introduce doses of compound 3g
Response 1: Doses of the tested compound have been introduced in the mentioned line.
Point 2: Line 101: action is the increase…
Response 2: "The" has been added to the sentence.
Point 3: Line 102: “…formalin is injected intradermally in one of the hind paws one hour after the administration of the tested substances.… : The authors explain the choice of this time distance
Response 3: Drug administration schedule is based on previous reports - a reference for the time interval has been added.
Point 4: Line 112: instead in Anti-inflammatory activity assay, the authors said that “Saline solution, diclofenac sodium and compound 3g (10, 20 and 40 mg/kg b.w.) are administered intraperitoneally immediately after the injection of carrageenan" without allowing a minimum distance between the test substance and the inflammatory agent. The authors explain this choice.
Moreover, the authors explain because the carrageenan solution was used in 1% instead of 2%.
Response 4: A reference has been added to explain the choice of % for the carrageenan solution. Drug administration schedule was selected based on the following - according to previous reports substances are administered 1hr after carrageenan and testing begins 1hr after that. Since our testing begins 2hr after carrageenan administration, our test substances were injected right after the carrageenan.
Point 5: Finally, the tables 2, 3 and 4 could be replaced with graphs.
Response 5: Since the reviewer is not insistent on changing tables into graphs, we have kept the tables in order to make the paper more cohesive.